# Assessment of Reactivity to the Administration of the mRNA Vaccine after Six Months of Observation

**DOI:** 10.3390/vaccines11020366

**Published:** 2023-02-06

**Authors:** Sebastian Slomka, Patrycja Zieba, Oskar Rosiak, Anna Piekarska

**Affiliations:** 1Department of Internal Medicine and Geriatry, Bieganski Regional Specialist Hospital, 91-347 Lodz, Poland; 2Department of Otolaryngology, Polish Mother’s Memorial Hospital Research Institute, 98-338 Lodz, Poland; 3Department of Infectious Diseases and Hepatology, Medical University of Łódź, 90-419 Lodz, Poland

**Keywords:** SARS-CoV-2, mRNA vaccine, IgG SRBD antibody, COVID-19

## Abstract

Background: The fast spread of the SARS-CoV-2 virus accelerated efforts to create an effective vaccine, and a novel mRNA vaccine was the first to appear effective. Scientific evidence regarding mRNA vaccination is limited; therefore, understanding how the immune system responds to an mRNA vaccine is critical. Our study was aimed at a long-term analysis of the presence and maintenance of the immune response using the chemiluminescent method by analyzing the level of IgG antibodies in vaccinated people who were and were not infected with the SARS-CoV-2 virus. Materials and methods: Healthcare workers with a history of COVID-19 or who were naïve to the infection were recruited for this study and administered two subsequent doses of the Comirnaty vaccine. IgG SRBD antibody levels were evaluated every month for six consecutive months using the chemiluminescent immunoassay (CLIA). Results: A total of 149 individuals were recruited for this study, 68 had evidence of past COVID-19 infection, with 63 exhibiting elevated IgG SRBD antibody levels at initial evaluation. Statistically significant differences were observed between COVID-19 convalescents and non-convalescents at all study time points, with the convalescent group consequently representing higher antibody levels. Conclusions: COVID-19 convalescents showed a stronger immune response to the vaccine after the first dose. This group exhibited higher IgG levels in all examinations during the observation period. The natural waning of antibody levels can be observed within six months. A booster vaccination may be required. No serious side effects were observed.

## 1. Introduction

The outbreak of the COVID-19 pandemic in 2019, caused by SARS-CoV-2, accelerated the introduction of novel mRNA vaccines. The concept of mRNA vaccines is based on the idea of using the genetic information itself, rather than the viral protein antigen, to produce viral proteins. Therefore, an mRNA vaccine is not a virus, but only information about how a virus protein has to be constructed. The first work on the possibility of using mRNA material as a carrier of genetic information for endogenous protein expression began in the early 1990s [1]. However, due to the complex process and the high production costs for synthetic RNA vectors, the first mRNA vaccines were not developed until 2008 [2], and their use as a vaccine was only approved by the WHO in 2016.

There are currently 119 coronavirus vaccines in Phase I, II, and III clinical trials, and approximately 50 have reached the final stages of testing. More than 75 vaccines are in the pre-clinical stage (as of 25.03.2022, according to a WHO report). At the time of writing, two mRNA vaccines (Comirnaty from Pfizer and Moderna) have been available in Poland since 28 December 2020. A third, CurVac, was withdrawn after efficacy at the end of the studies was estimated at 47%. The first vaccine approved for use in the EU was the Comirnaty vaccine.

Immune surveillance by the measurement of antigen-specific antibody levels is fundamental to assessing vaccine efficacy. The immune system’s response to infection with SARS-CoV-2 is similar to that of other viruses, and there are two types of response: non-specific and specific [3]. Both processes run parallel. In addition to the cellular response, which is the non-specific mechanism, a humoral response, which is related to the production of antibodies, is the specific body response to infection. The helper-T-lymphocytes are responsible for this, as they support the humoral response by activating the B-lymphocytes to form antibodies. There are five classes of antibodies: IgM, IgG, IgA, IgE, and IgD. In primary infection by SARS-CoV-2, the IgA and IgM antibodies are the main class of antibodies responsible for deterring the infection, while IgG antibodies have the function of preventing re-infection [4]. The defensive response can be summarized in two stages. First, the IgM antibodies neutralize the virus by attaching to the S protein, thereby primarily blocking infection of cells expressing the human angiotensin 2 converting enzyme (hACE2) [5]. Then, the infected cells are eliminated by cellular antibodies and antibody-dependent phagocytosis [4]. We assess post-vaccination immunity using serological tests based on the determination of IgG anti-S antibody levels. The evaluation of other classes of antibodies does not play a role in vaccine protection indicating immunity; it can only serve as a marker for a fresh infection. IgG antibody levels should be tested at the earliest 7–15 days after the second vaccination.

As an envelope virus, the coronavirus contains four structural proteins: the spike (S) protein, the membrane (M) protein, the envelope (E) protein, and the nucleocapsid (N) protein. Among these proteins, the S and N proteins are the most immunogenic, in particular, the S protein, which is the main protective antigen responsible for the strong reaction of neutralizing antibodies (nABbs). It also plays an important role in the adhesion of the virus to the host cell, fusion, entry, and transmission. The S protein consists of two subunits: S1, for binding the virus to the receptor, and S2, for cell membrane fusion. One of the domains of the S1 subunit is the receptor-binding domain (RBD), which interacts directly with the host receptorhACE2 (the human angiotensin-2 converting enzyme). This element of the virus (the RBD domain) is a key target for neutralizing antibodies (nAbs) by inhibiting the interaction between RBD and ACE2.

Clinical trials of SARS-CoV-2 vaccines did not include long-term follow-up due to the rapid introduction of this method in infection prevention during the pandemic period.

First of all, it is interesting how long the protective antibodies persist. Our previous observations suggest that young females showed the highest levels of antibodies, but also developed the most adverse events after vaccination. Furthermore, in convalescents, we observed worse tolerance of mRNA vaccines (more side effects). This knowledge is essential to determine the vaccination schedule and whether the convalescents should be vaccinated in the same mode as people who have not suffered from SARS-CoV-2 infection.

The purpose of the project is to assess the effect of immune stimulation in a group of people with positive and negative COVID-19 status who have been vaccinated with the mRNA vaccine. The main research hypothesis is that the level of persistence of specific IgG anti-SARS-CoV-2 (IgG SRBD) antibodies is dependent on previous exposure to the virus. The secondary aim is to assess whether the post-vaccination side effects are different between the convalescent and non-convalescent populations. The mRNA vaccine evaluated in this project was the Comirnaty vaccine as it was the first approved vaccine in the EU and was available in Poland when this project began. Per government regulation, the first to receive the vaccine were the healthcare professionals, and that is why we conducted our study among medical workers whose health and ability to work were crucial during the pandemic.

## 2. Materials and Methods

The research protocol was approved by the Bioethics Commission of the Medical University of Lodz (RNN/29/21KE). All subjects provided written informed consent. All clinical investigation was conducted according to the principles expressed in the Declaration of Helsinki.

Patients were recruited from medical and non-medical workers of USK No. 1 of the Medical University of Lodz, as healthcare workers were the first group to be vaccinated with the mRNA vaccine in European Union countries. Subjects who met all of the inclusion criteria and none of the exclusion criteria were included in the study. The study was performed from January to August 2021. Inclusion criteria were: adult patients aged 18–80 years, no contraindications for the Comirnaty vaccine (fever or symptoms of infections on the day of vaccination), and willingness to attend follow-up visits. Exclusion criteria were pregnancy, upper respiratory tract infection in 14 days preceding the vaccination, inability to provide informed consent to participate in the study, and a history of asthma, diabetes, or chronic obstructive pulmonary disease.

Upon meeting the inclusion criteria and none of the exclusion criteria, informed consent to participate in the study was obtained from the patients. The blood sampling followed the protocol detailed in Table 1. Blood samples were drawn upon the first visit (qualification for vaccination and administration of first dose), then on the day of the second dose of vaccine (21st–28th day), and then every 30 days for 6 months.

Each study participant received three questionnaires to complete: a history of SARS-CoV-2 infection, and vaccine side effects after the first and second doses. There are 14 symptoms in the questionnaires: low-grade fever, fever, cough, breathlessness, muscle aches, headache, weakness, rhinitis, stuffy nose, nausea, vomiting, diarrhea, concentration disorders, and disorders in taste and smell. The participants rated each symptom on a 0–3 scale based on subjective opinions. The researchers then assigned the participant one of the categories: 0 points–asymptomatic course; 1–14 points—mild course; 15–25 points—moderate course; and >25 points—severe course.

The surveys were aimed at selecting a group of people with asymptomatic SARS-CoV-2 infection and studying the relationship between the severity of the disease and the amount of vaccine antibodies.

## 3. Description of Laboratory Methodology

The study used two types of tests to analyze the occurrence of IgG antibodies. The first is the 2019-nCoV IgG/IgM (CLIA) chemiluminescence test, which should rule out or confirm infection with SARS-CoV-2. The test mentioned above is based on the N protein, which is responsible for the formation of the SARS-CoV-2 nucleocapsid. In vaccinated individuals (not previously exposed to SARS-CoV-2), antibodies to the coronavirus N protein are not synthesized, and the presence of this antibody has been shown not to be relevant for virus neutralization, as is the case with SARS-CoV-2-RBD antibodies. IgG (CLIA) is used to quantify SRBD IgG antibodies to SARS-CoV-2 in vitro in human serum or plasma. This is due to the fact that the S protein is the main antigen used in SARS-CoV-2 vaccines. Therefore, only antibodies to SARS-CoV-2-SRBD IgG in human serum or plasma correlate with the protective response of the immune system in vaccinated individuals while simultaneously reflecting immunity in the population. We used the MAGLUMI^®^ SARS-CoV-2 SRBD IgG CLIA (New Industries Biomedical Engineering Co., Ltd. [Snibe], Shenzhen, China) which was granted Emergency Use Authorization by the US Food and Drug Administration, anti-NCP during the initial months of the pandemic followed by anti-SRBD later in 2020.

## 4. Statistical Analysis

Data were stored on a computer and analyzed using Statistica 13.1 Software (TIBCO Software Inc, Palo Alto, CA, USA). Nominal variables were compared between the groups in contingency tables using the chi-squared Yates test and Fisher’s exact test for cell counts lower than 5. Continuous variables were tested for normality of distribution using the Shapiro–Wilk test. An alpha level of 0.05 was established. For non-normally distributed variables, the median and interquartile range (IQR) was analyzed, while for normally distributed variables, the average and standard deviation (SD) were analyzed. A two-way Wilcoxon test was used for comparing paired samples, the U-Mann–Whitney test was used for non-paired comparisons, while for multiple pairwise group comparisons, the Friedman ANOVA was utilized; the results were then followed by a Nemenyi post hoc test. The criterion of statistical significance *p* < 0.05 was used in the statistical analyses.

## 5. Results

### 5.1. Study Group

It must be noted that the study commenced in January 2021 and finished in August 2021 and only one virus variant was present in the Polish population at the time. The subjects of this study were among the first to receive the vaccine in the general population.

The study population included 149 patients: 27 males (18.12%) and 122 females (81.87%). The age distribution was non-normal. The median age was 46 years (IQR 18); the youngest participant was 23 years old; and the oldest was 69 years old, which is the pre-retirement maximum for healthcare workers in Poland (Table 2). 

The median age of the females was 48 years (IQR 19), and the median age of the males was 38 years (IQR 19). There was a difference in age between the genders (U-Mann–Whitney test *p* = 0.043).

In the patient-reported questionnaires, 68 people declared COVID-19 positive status in the past months (45.63%). Of these individuals, five people did not present an IgG antibody count above 1.0 at the Day 1 initial assessment (7.35%). The remaining population of 81 patients did not declare a past COVID-19 infection (54.36%); 23 individuals in this group (28.4%) had an elevated initial IgG antibody count and were considered to be asymptomatic COVID-19 convalescents.

The total number of COVID-19 convalescents considered to be patients who presented with an initial elevated IgG count was 86 individuals, while the remaining 63 patients did not present an IgG count over 1.0 at initial bloodwork. There was no difference in age between the COVID-19 convalescents (median 46.5 IQR 18.0) and those who had not experienced COVID-19 (median 45.0 IQR 21.0).

### 5.2. Serum Antibody Level Analysis in Time Points

Serum IgG antibody levels were analyzed in time, and there was a significant change in antibody levels for both groups (convalescents and non-convalescents) Figure 1.

Age was not found to be a significant factor influencing the IgG synthesis among the individuals included in this study.

Comparing the serum IgG SRBD levels in the convalescent and non-convalescent groups, there is a rise in antibody levels at the second assessment, which is different between the groups. The convalescent group noted a three-fold increase in antibody levels, and at the second assessment (after the first dose of the vaccine), they had a significantly higher level than the non-convalescent group at the second assessment (*p* < 0.001). However, the non-convalescent group antibody level increased almost 54 times compared to the initial assessment. A gradual decrease in antibody levels was observed over time, although at all time points, the convalescent group retained a higher SRBD level. This is summarized in Table 3.

### 5.3. Patient-Reported Post-Vaccine Side Effects

Analyzing the occurrence and severity of the post-vaccine reactions described in questionnaires at Visit 2 and 3, there is a visible difference in the severity of symptoms reported by the convalescent group, which was 3.06 times more likely to report mild to moderate vaccine-related side effects (OR 3.06 95%CI 1,54; 6,10) after the second vaccine dose. No such observations were made for side effects after the first dose for either group. None of the participants reported symptoms that were classified as “severe,” only three participants reported “moderate” symptoms after the first dose of the vaccine, and four patients reported “moderate” symptoms after the second dose. This is summarized in Table 4.

### 5.4. Symptomatic and Asymptomatic COVID-19 Convalescents

We analyzed whether the group of convalescents that presented symptoms suggestive of COVID-19 infection and presented with elevated initial IgG serum concentration (n = 63) had a difference in serum SRBD concentrations over the time of the study compared to the group of asymptomatic COVID-19 convalescents (n = 23). The only difference could be seen upon the initial visit with the symptomatic group presenting two times higher IgG levels (*p* = 0.029, U-Mann–Whitney non-parametric test).

## 6. Discussion

The study showed significant differences in response to the vaccine between the group that had an infection and the group that had no data on SARS-CoV-2 infection. A significant difference in response to the second dose of vaccine has been demonstrated in both antibody concentration and response to vaccination in terms of vaccination side effects. Based on these results, it can be expected that the group with both symptomatic and asymptomatic COVID-19 infection may receive a lower vaccination dose.

These observations are in line with other studies on the subject. A study by Harika Oyku Dinc et al. [5], who assessed health professionals in Turkey, showed that the antibody response is significantly higher in those with prior history of COVID-19 than in the infection-naïve group. Those with a prior history of COVID-19 developed significantly higher antibody responses after the first dose of vaccine (96.4% vs. 48%), yet the antibody development rates after the second dose were similar (99% vs. 100%). Hence, there was a significant decrease in the median antibody titers of HCWs with hypertension (488.9 vs. 731.5) without prior history of infection. There was no difference between the two groups when evaluated in terms of other comorbid diseases and blood groups. The researchers hypothesized that the second dose of vaccine is of little benefit in the COVID-19 convalescent group, and a single dose may be sufficient with prior COVID-19 testing. However, with the emergence of new variants of the virus, these observations should be re-evaluated.

However, the results of comparative studies on the response to vaccination in convalescent and non-ill people are not unambiguous. A study by Ibarrondo et al. [6] showed that people without prior COVID-19 infection achieved similar antibody levels to those of a mild natural infection after the first infection. In contrast, in the convalescent group, one dose boosted the antibody levels to those observed in the end phase of a severe infection, increasing no further after the second dose. The authors also noted a natural waning of the antibodies in the following 90 days, underlining the possible need for booster vaccinations. To date, no lower limit of antibody titers necessary to protect patients from possible infection has been established. In our study, no signs of infection or a positive SARS-CoV-2 test were observed in the study group during the six-month follow-up period. Given the close contact with COVID-19 patients in the healthcare worker group, it should be assumed that the values generated are sufficient to maintain the patient for at least six months. However, this contradicts the study by Ibarrondo et al. [7], which showed a sharp decline in IgG antibodies within 90 days. A similar trend was observed in the group studied by a team of scientists from Chile, where after 138 days 14.2% of volunteers (3/21) did not have anti-SARS-CoV-2 IgG [8]. A different study by Wang Y et al. [9] also confirmed that antibody concentrations remained relatively high in both groups over a period of six months, and antibody titers reached maximum levels after the disease and persisted significantly longer, as shown by Choe et al. [10] and Dehgani-Mobaraki et al. [11]. The observed kinetics of a decrease in antibody levels suggest the need for regular re-vaccination, which is confirmed by available studies at other sites [6]. Without adjusting the standard doses of the vaccine, the convalescent group may additionally receive a booster dose about two months after the resolution of symptoms. Furthermore, our study confirmed the high efficacy and usefulness of the CLIA tests [12]; their sensitivity in the meta-analysis of Bastos et al. [13] was 97.8% (compared to 84.3% of ELISA tests).

The course of a SARS-CoV-2 infection is largely unpredictable, and over the first months of the pandemic, it became clear that an effective boost to natural defenses against the virus was required. Since natural immunity after the disease and prevention of infection was far from sufficient, vaccines were developed, a vaccination schedule developed, and their safety profile assessed [14]. In our study, we looked at one occupational group that was one of the first to get the vaccine.

As many as 99.2% (1 person was a non-responder) of the subjects responded positively and developed antibodies. Pre-vaccination antibody concentrations were low, even in severely infected individuals, but the immune response to vaccination was significantly higher. In this way, we can conclude that the entire population needs vaccinations and that they are effective. There was only a problem assigning populations to the above groups with SARS-CoV-2 patients who did not have antibody formation at baseline, with a rate of about 7% in the submitted study. Another point is the rating of people who had an asymptomatic infection—28.4%. Determining baseline antibodies prior to vaccination and then allocating patients to one of the groups is optimal [3,15]. An additional positive finding from the study is that no side effects are to be expected after vaccination, as none of the subjects had serious adverse events, which would not be expected in patients with COVID-19.

Slightly different trends in response to vaccine doses were observed in this study. Overall, fewer side effects occurred after the second dose of the vaccine, and this difference was significantly greater in the group of people who did not suffer from SARS-CoV-2. In large population studies conducted in Israel [16] and the United Kingdom [17], this trend was reversed, which may be related to the much larger sample population.

This study has several limitations. Among them is the small sample size, limiting participants to those under 70 years of age, and the significant predominance of women. Another limitation is that the study population is exclusive to healthcare professionals. Despite the high probability of exposure to the virus, they are also well-equipped and educated in protective behavior and personal protective equipment. Other types of vaccines, including vector vaccines, should be evaluated in this manner and compared with the mRNA vaccine results. A longer follow-up should also be considered as, currently, the virus has several new variants, which might render the vaccine less effective in longer observation. Neutralizing activity was not assessed due to technological limitations at the beginning of the study. Analyzing the neutralizing antibodies could provide additional scientific value. The limitation of this study is also the observation of only one type of vaccine and the humoral response based on the assessment of total antibodies.

## 7. Conclusions

The results of the study clearly show the efficacy of Comirnaty (tozinameran/BNT162b2) in maintaining effective protection against the virus in six months of observation, with a high response to the vaccination in all individuals. We observed high antibody levels after the first dose of the vaccine in the convalescent group, which is significantly different from the naïve, unexposed population of patients, who required a second dose to achieve a similar response. The natural waning of antibodies can be observed in both groups, with second doses probably required after six months of observation. However, it is advisable to consider a lower dose or just one dose for people with COVID-19 [18]. In addition, we found that a booster dose should be given earlier to people without a history of infection to ensure optimal protection against SARS-CoV-2 infection. No severe side effects after the vaccination were observed.

## Figures and Tables

**Figure 1 vaccines-11-00366-f001:**
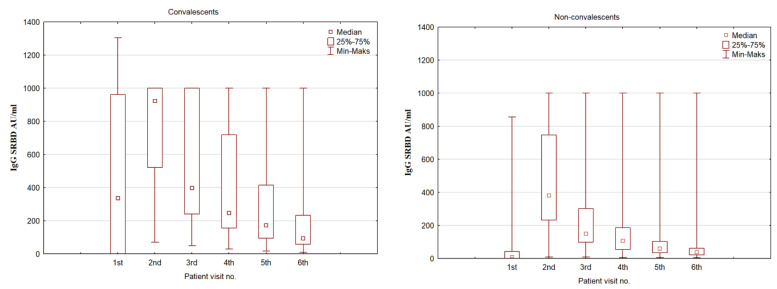
A box plot representing the IgG SRBD serum concentration in COVID-19 convalescents and non-convalescents during the time of the study at predestined time intervals.

**Table 1 vaccines-11-00366-t001:** Table detailing the timeline of procedures and laboratory workup in this study.

Procedures/Time	1st day	21st–28th day	60th day	90th day	120th day	150th day	180th day
Signing the consent	+						
Analysis of inclusion and exclusion criteria	+						
Medical history, questionnaire	+	+	+				+
Blood sampling	+	+	+	+	+	+	+

**Table 2 vaccines-11-00366-t002:** General characteristics of the study population.

Age	Median 46, Min. 23, Max. 69
Gender	27 male (18.12%)/122 female (81.87%)
Race	N = 149 Caucasian (100%)
Comorbidities	Hypertension N = 9 (6%) Metabolic syndrome N = 12 (8%) Hipercholesterolemia N = 12(8%)

**Table 3 vaccines-11-00366-t003:** The serum levels of SRBD IgG antibodies in COVID-19 convalescents and non-convalescents at baseline and follow-up visits.

Visit No.	Convalescents (n = 86) IgG SRBD AU/mL		Non-Convalescents (n = 63) IgG SRBD AU/mL		U-Mann–Whitney Test	
	Median	IQR	Median	IQR	*p*-Value	Z Score
1st (at 2nd dose)	335.85	961.5	7.878	41.53	<0.001	3.686
2nd (1-month FU)	922.8	478.9	381.85	515.5	<0.001	5.670
3rd (2-month FU)	397.8	760.3	150.85	203.92	<0.001	6.093
4th (3-month FU)	245.9	561.8	106.1	130.38	<0.001	5.416
5th (4-month FU)	172.35	320.56	59.55	68.65	<0.001	5.689
6th (5-month FU)	94.59	174.12	37.115	41.4	<0.001	5.672

FU—Follow-up; IGR—Interquartile range.

**Table 4 vaccines-11-00366-t004:** Reported post-vaccine symptom severity across the COVID-19 convalescent and non-convalescent groups.

	Symptom Severity After 1st Dose			Symptom Severity after 2nd Dose		
Symptom Severity	None	Mild	Moderate	None	Mild	Moderate
Convalescents (n = 86)	32 (37.2%)	53 (61.6%)	1 (1.2%)	37 (43%)	47 (54.7%)	2 (2.33%)
Non-convalescents (n = 63)	26 (41.3%)	35 (55.6%)	2 (3.2%)	44 (69.8%)	17 (27%)	2 (3.2%)
Fischer’s exact test with Freeman–Halton extension for 3 × 2 contingency tables	*p* = 0.5			*p* = 0.002		

## Data Availability

The data presented in this study are available on request from the corresponding author. The data are not publicly available due to privacy policy (principles of personal data protection in the EU).

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
