# Peer review of "Assessment of Reactivity to the Administration of the mRNA Vaccine after Six Months of Observation"

_vaccines, 2023, doi:10.3390/vaccines11020366_

Round 1

Reviewer 1 Report

Slomka and colleagues examined antibody levels to sRBD SARS-CoV-2 in first responders as to if there were differences between those that coalescent or not. Much of the introduction needs to be refocused and contained negative or stigmatizing language. Many studies have looked at antibody levels such as this study, and while perhaps novel in the author's specific setting, it confirms data at most, and could be served as a technical briefing. A participant demographics table for stratification would be quite helpful. It is encouraging that the authors demonstrate similar results to what other countries have reported as well. It is just in the opinion of this reviewer that more is said and opined than needed.

Author Response

Response to reviewer 1

1.Thank you for this suggestion, we have reviewed the introduction section of the manuscript. The changes are now visible in the revised version. We have removed data from other viruses and some sentences that were too opinionated. Please find the updated Introduction section in the attached manuscript file.

  1. The second table in the revised version now contains demographic data.

Reviewer 2 Report

This article shows the behavior and production of antibodies after complete doses of mRNA vaccines. To date, a series of reviews of papers demonstrating the impact of vaccination long IgG production have been described.

Although the data analysis is interesting and includes a convalescent population compared to the non-infected population, I believe that a series of improvements are necessary for this work.
The introduction should not include data from other viruses, they are unnecessary. The mRNA vaccines used in the country, and included in this report, must be described in the introduction and demographic characterization. Likewise, it is necessary to describe at least briefly the epidemiological data in which the present analysis is carried out, which virus variants were present, and also the vaccinated population, among others.
Material and methods should be summarized (sometimes it looks like an introduction). The methodology must correspond to the description of the methods used and the reagents used.

The antibody kit corresponds to a commercial kit to detect total antibodies, this does not account for their neutralizing activity. This point should be considered for discussion.
The type of graph presented, in my opinion, is not the best way to present the data. I suggest a scattered plot graph, in which it is possible to analyze all the data.
Is it possible to determine particular curves for each patient to see their behavior (see 10.3389/fimmu.2021.766278 vaccine.)? I think it would be interesting to be able to observe the dates in these same curves or times in which the patients were infected and see if there is a different behavior. Are mRNA vaccines specified, what are they, and is there any behavior associated with the different vaccines?
Although the ages and number of participants are restricted, I suggest analyzing when possible whether there are differences by setting a cut-off point between adults and older.
A major point of interest may be the comparison with other vaccines and the effect of antibodies as in the case of the Sinovac 10.3389/fimmu.2021.766278 vaccine.
Considering that post-vaccination symptoms stand out in the description of results, this should at least be included in the discussion and compared with other works or what is described by the manufacturers themselves based on published clinical studies.
The limitations must include the lack of evaluation of cellular parameters and that they do not correspond to neutralizing antibodies but total antibodies. It should also be included in the discussion.

Author Response

Response to reviewer 2.

  1. The introduction should not include data from other viruses, they are unnecessary.

Ad. 1 Thank you for this suggestion, we have reviewed the introduction section of the manuscript. The changes are now visible in the revised version. We have removed data from other viruses and some sentences that were too opinionated. Please find the updated Introduction section in the attached manuscript file.

  1. The mRNA vaccines used in the country, and included in this report, must be described in the introduction and demographic characterization.

Ad. 2 Since this study was commenced at the time when only one vaccine (Comirnaty) was available only this mRNA vaccine was included in the report. This is now outlined at the end of the introduction paragraph “The mRNA vaccine evaluated in this project was the Comirnaty vaccine as it was the first approved vaccine in the EU and available in Poland when this project begun. Per government regulation first to receive the vaccine were the healthcare professionals.”

The study demographic is characterized in a new Table 2.

  1. Likewise, it is necessary to describe at least briefly the epidemiological data in which the present analysis is carried out, which virus variants were present, and also the vaccinated population, among others.

Ad. 3 This study was conducted on the first patients to receive the first available vaccine, therefore no other vaccinated population was available at the beginning of the trial. Likewise, at this point only one virus variant was present in the population. This is now outlined in the introduction section and in the results section of the manuscript in 6.1 “It must be noted that the study was commenced in January 2021 and finished in August 2021, only one virus variant was present in the Polish population at the time.The subjects of this study were among the first to receive the vaccine in the general population. “

  1. Material and methods should be summarized (sometimes it looks like an introduction). The methodology must correspond to the description of the methods used and the reagents used.

Ad 4. Thank you for this suggestion, we have modified the laboratory methodology subsection to be more brief.

  1. The antibody kit corresponds to a commercial kit to detect total antibodies, this does not account for their neutralizing activity. This point should be considered for discussion.

Ad 5. In a parallel study comparing two types of vaccines (mRNA vs vector) we used additional kits to assess the level of neutralizing antibodies. However, this is a completely separate research material.

Neutrailizing antibodies were not analyzed in this study, this is now included in the limitations section of the manuscript. “Neutralizing activity was not assessed due to technological limitations at the beginning of the study. Analyzing the neutralizing antibodies could provide additional scientific value.”

  1. The type of graph presented, in my opinion, is not the best way to present the data. I suggest a scattered plot graph, in which it is possible to analyze all the data.

Ad. 6 As per reviewers request we have changed the graph to scattered plot, however we believe that median and a line connecting the median points helps in visualizing the data, therefore it is also included in the new graphs.

  1. Is it possible to determine particular curves for each patient to see their behavior (see 10.3389/fimmu.2021.766278 vaccine.)? I think it would be interesting to be able to observe the dates in these same curves or times in which the patients were infected and see if there is a different behavior.

AD 7. We believe that this graph would not be readable as the study population is 149 individuals and there are 6 time points at which the antibodies were assessed. We include these graphs here for reviewers evaluation, however we believe they do not add value to the manuscript.

  1. Are mRNA vaccines specified, what are they, and is there any behavior associated with the different vaccines?

Ad. 8 At the time of the study only one mRNA vaccine was available to patients in Poland therefore only the Comirnaty vaccine is the subject of this study.

  1. Although the ages and number of participants are restricted, I suggest analyzing when possible whether there are differences by setting a cut-off point between adults and older.

Ad  9. We have analyzed the age using Weight of Effect (WoE) and split the group into three age categories 23-35;36-50;51-69. A Kruskall-Wallis ANOVA analysis of the three age groups at 6 time points (blood tests) revealed no significant differences between the groups

1st IgG test p =0.116

2nd IgG test p= 0.23

3rd IgG test p = 0.211

4th IgG test p= 0.22

5th IgG test p= 0.194

6th IgG test p=0.426

We have now included a short statement in the results section under 6.2 “Age was not found to be a significant factor influencing the IgG synthesis between the individuals included in this study.”

  1. A major point of interest may be the comparison with other vaccines and the effect of antibodies as in the case of the Sinovac 10.3389/fimmu.2021.766278 vaccine.

Ad 10. We added:

“A similar trend was observed in the group studied by a team of scientists from Chile, where after 138 days 14.2% of volunteers (3/21) did not have anti-SARS-CoV-2 IgG [8]”

“The limitation of this study is also the observation of only one type of vaccine and the humoral response based on the assessment of total antibodies.”

  1. Considering that post-vaccination symptoms stand out in the description of results, this should at least be included in the discussion and compared with other works or what is described by the manufacturers themselves based on published clinical studies.
    The limitations must include the lack of evaluation of cellular parameters and that they do not correspond to neutralizing antibodies but total antibodies. It should also be included in the discussion.

Ad 11. As suggested, we have added the following excerpt to the discussion:

“Slightly different trends in response to vaccine doses were observed in this study. Overall, fewer side effects occurred after the second dose of the vaccine, and this difference was significantly greater in the group of people who did not suffer from SARS-CoV-2. In large population studies conducted in Israel[22] and the United Kingdom[23], this trend was reversed, which may be related to the much larger sample population.”

Round 2

Reviewer 1 Report

Slomka et al have made improvements in the overall article, but there are still some concerns. While it is understandable that safety and efficacy might be needed to be supported within individual countries and communities, the data as presented re-iterates findings that have been already been established. 

1. Line 61, "body cells" is vague. Could state primarily cells that express hACE2 on the surface, since you describe the mechanism in the next paragraph.

2. Line 68, 'Spike' not spine for S gene

3. Line 82: IgGp/SARS-CoV2, what does the 'p' refer to? Is this just a typo?

4. General, there is inconsistencies between SARS-CoV2 and SARS-COV-2, etc.

5. It is noted that there was a significant difference between males and females in the study. Are the authors able to weigh these results to account for the major imbalance?

6. There were several font changes throughout the manuscript and spelling/abbreviation errors.

7. Lines 218-223: This has been established countless times by many groups in the past two years. 

8. Lines 227-229: This is not in-line with what the current data suggests. Even in those that convalescence (eventually), boosting is required, especially since new variants continue to emerge. The study does not look at times past 5-6 months, and the lack of antibodies continues to wane to near undetectable levels >1 year.

Author Response

I am sending the manuscript corrected according to the reviewers' suggestions and supplemented with a graph.

Reviewer 2 Report

The work improved a lot, I think that the way of presenting the data can be staggered, allowing you to see the density at each point. Likewise, given that it is an individual follow-up, I reiterate my request to incorporate (even as a supplementary figure) a graph that joins each patient with lines, this would allow observing the changes in the levels of each one and determining if there are populations with different capacity for the reply.

Author Response

(The authors gave the same response as above.)

Round 3

Reviewer 1 Report

The authors have satisfied the concerns at this point in time.